# ROYAL SOCIETY
# OPEN SCIENCE

biophysics/biomathematics/biocomplexity

constructal law, homeostasis, macrotermes, termite mound, thermal optimization, thermoregulation

**Author for correspondence:**
Neda Yaghoobian
e-mail: nyaghoobian@eng.famu.fsu.edu

# How the thermal environment shapes the structure of termite mounds

Tadeu Mendonca Fagundes, Juan Carlos Ordonez and Neda Yaghoobian

FAMU-FSU College of Engineering, Florida State University, Tallahassee, FL, USA

TMF, 0000-0002-8874-5255; NY, 0000-0002-3399-8650

A computational model has been developed to predict the role of environment in the forms and functions of termite mounds. The proposed model considers the most relevant forces involved in the heat transfer process of termite mounds, while also reflecting their gas-exchange function. The method adopts a system configuration procedure to determine thermally optimized mound structures. The model successfully predicts the main architectural characteristics of typical *Macrotermes michaelseni* mounds for the environmental conditions they live in. The results indicate that the mound superstructure and internal condition strongly depend on the combined effect of environmental forces. It is noted that mounds being exposed to higher solar irradiances develop intricate lateral channels, inside, and taller and more pronounced spire tilt towards the Sun, outside. It is also found that the mounds' spire tilt angle depends on the geographical location, following the local average solar zenith angle for strong irradiances. Although wind does not influence the overall over-ground mound shape, it significantly affects the mound internal condition. The results of this study resonate with what is seen in nature. The proposed approach provides a broader view of the factors that are effective in the form and function of a naturally made structure.

## 1. Introduction

In nature, some animals build structures to shelter themselves against environmental and physical threats. In some species, the function of these natural structures goes beyond a roof-over-head, as they are means for regulating temperature and moisture, and for gas exchange. The odd-shaped structures that mound-building termites create are famously known for providing both sheltering and environmental regulating functions [1–6]. Historically, the internal environment regulation and ventilation aspect of termite mounds have received more

attention than their morphology and inspired most of the past studies in the field. These studies led to several theories relating the mound gas-exchange process to the metabolism-induced buoyant forces [7–9], forces due to the interactions between the mound superstructure and the atmospheric turbulent wind flow [5], or forces induced by the diurnally variable mound surface temperature [10–14].

It is noted that the forms and physical characteristics of the mounds, ranging from small domes to massive cone-, cathedral- and wedge-shaped structures, depend on the local climate and available materials. However, no definite correlation between termite species and mound shapes has been identified [15]. As an example, Korb & Linsenmair [16] studied two mounds of *Macrotermes bellicosus* termites with different structural characteristics in one geographical location. The one built in hot, open savannah was described as a more complex mound with thinner walls and a larger surface area compared to that in a cooler shaded area. Furthermore, Kooyman & Onck [17] reported that mounds of *Pseudacanthotermes spiniger* termites in a hot environment can be as twice as tall as those built in a nearby colder condition, while Turner [18] noted that mounds of *Macrotermes michaelseni* termites located under tree shadows tend to be more upright than those directly exposed to the Sun. The large diversities/similarities seen among termite mounds in one/different geographical locations and species have made a comparative study challenging. Recently, Claggett *et al.* [15,19] began an effort to use existing databases to extract the relationship between the environmental conditions and the structural form of termite mounds—a correlation that has, historically, been understudied [15]. Their study concluded that, in addition to the environmental conditions, the soil composition and property also play a significant part in the resulting mound shape.

In this study, we use an engineering point of view to address the connection between the structural form of termite mounds and environmental forces. The question to be answered is 'how does the mound's structure adapt to its local environment to provide a favourable living condition for its termites?' The study implements a system configuration procedure, supported by the Constructal Law, to determine an optimized mound structure. The Constructal Law states that for a finite-size flow system to persist in time (to live), its configuration must evolve in a way that provides easier access to the currents that flow through it [20,21]. As long as a system has the freedom to alter its form, it will evolve its configuration in time to allow for better access to the fluxes that flow through it, while minimizing the holistic resistance and losses in the system.

The study uses the structural characteristics of the mounds of *M. michaelseni* termites as the base since available information and literature on the physical features of these mounds are far richer compared to the other termite species. The aim is to investigate whether environmental (wind and solar irradiance) and metabolism-induced forces involved in the heat transfer process of termite mounds affect their structural configurations and if yes, in which direction. The methodology developed here could, in principle, be applied to any termite mound and the overall approach could be extended to other animal-built structures, where thermodynamics and heat transfer play important roles in their functions.

The paper is structured as follows: Section 2 reviews the characteristics of the *M. michaelseni* termite mounds, focusing on the information used in the study to create the heat transfer model described in §3. Section 4 validates the model for different structural set-ups. Section 5 discusses the effect of different environmental conditions (including solar irradiance, solar zenith angle and wind), geometric features and porosity on the mound architecture. Finally, §6 elucidates the conclusions of the analysis based on the results of the current study and previous observations.

## 2. Characteristics of the *Macrotermes michaelseni* termite mounds

A comprehensive study of the physical characteristics of the *M. michaelseni* mounds was conducted by Turner [18] over 303 mounds in northern Namibia (16°4.50′ E, 19°59.05′ S). The architecture of these mounds can be characterized by a cone-shaped base, topped by a thin inclined cylindrical spire that is about twice the height of the base (figure 1).[1] The outwash pediment, seen at the base, is a product of erosion off the mound structure. Turner reported that the mounds were inclined towards the north, at an angle equal to the local Sun's average zenith angle, or 'the direction of greatest warming by insolation' [18].

In addition, Turner's study indicated that the *M. michaelseni* mounds are, typically, around 2.1 m in height, with a base diameter of 2.4 m, and half-length diameter of 1 m, while the outwash is extended

---

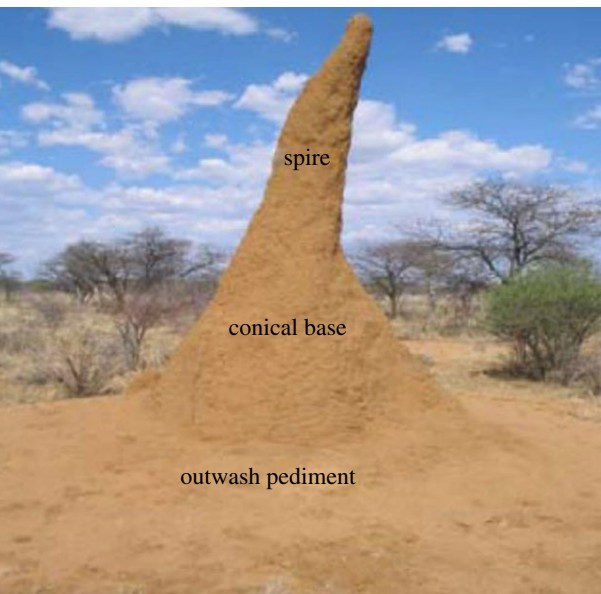

**Figure 1.** External structure of *M. michaelseni* termite mound outside Otjiwarongo, Namibia (the picture is adopted from [22]).

even further. The south face of the mound is 11.5% longer than the north face, reflecting the mounds' northward tilt of approximately 19°. Internally, *M. michaelseni* mounds are perforated by a complex network of air channels of different sizes. These channels converge at the centre of the structure to create a larger vertical passageway extending through the height of the mound. Typically, 18% of the mound body is composed of a network of tunnels that can be categorized into three distinctive types: a vertically oriented central chimney that extends from a subterranean nest to the top of the mound, surface conduits located under a thin porous covering and lateral channels that connect the surface conduits to the central chimney.

One important characteristic of *M. michaelseni* mounds is that their over-ground structure is empty and uninhabited, and termites live in an underground spherical-shape nest with an average diameter of 1.5–2 m (e.g. [7]). Termites found in the over-ground part of the mound are there either for repairing the structure or for defending the colony against intruding predators [7]. The metabolic heat generated within the nest is estimated to be 5–100 W, depending on the size of the colony [23].

There is a considerable diversity of mound shapes among *M. michaelseni* species. Most commonly, the variation comes in how distinct the spire is compared to the conical base [18]. However, for all cases, the mound is constructed primarily from quartz particles cemented together with a mixture of clay minerals and saliva [24], yielding a sturdy material that allows the mounds to endure harsh environments. The thermophysical properties of this specific composition were analysed in Oluyimika & Mijinyawa Yahaya [25].

## 3. Model description and simulation set-up

For the purposes of this study, a two-dimensional (2D) system that accounts for the main features of the mound of *M. michaelseni* is considered (figure 2). The justification for the utilisation of a 2D model is discussed in appendix A. The mound spire is modelled by a rectangular body with a width of $L'$ (m), funnelling angle of $\alpha$ and an inclination angle of $\gamma$. The conical base of the mound is represented by a trapezoid with a height of $H'$ (m) and lengths of $L$ (m) at the bottom and $L'/\cos(\gamma)$ (m) at the top. The mound total height is set to $H$ (m). The central chimney, which extends from the base of the model (the location of the underground nest) to the top of the spire, is represented by a highly conductive material (hereinafter, high conductivity pathway (HCP)), with a width of $D_0$ (m). This HCP has two lateral branches of the width of $D_1$ (m), representing the mound lateral conduits, which protrude from the central chimney at height $H_1$ (m). The lateral channels on both sides of the chimney were represented by the same height rather than being introduced independently. This simplification towards improving computational efficiency was imposed as our preliminary results indicated no sensitivity to this additional complexity in the model.

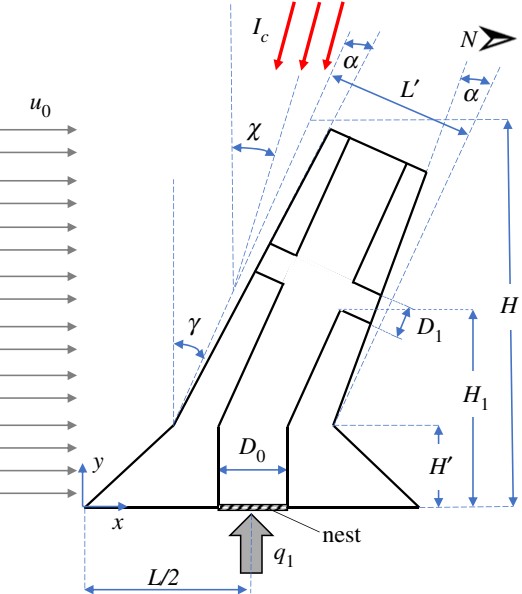

**Figure 2.** Simplified configuration of typical *M. michaelseni* termite mound with internal channels used in the model. $q_1$ and $I_c$, respectively, represent the metabolic heat rate from the nest (illustrated with a hatched patch) and solar irradiance.

The mound model is subjected to (i) solar irradiance $I_c$ (W m$^{-2}$) at a zenith angle of $\chi$, (ii) a heat rate of $q_1$ (W) imposed at the nest (located at the centre of the mound's base), representing the colony's metabolic heat, and (iii) the ambient wind. Diffuse solar radiation is considered uniform over the mound surfaces, while the amount of direct radiation depends on the angle between the incoming radiation and the outward normal to the surface. Assuming that the only mechanism to regulate the mound's internal environment is diffusion, we investigated the mound configurations that best diffuse the heat generated in the subterranean nest. The weak convective process in the mound conduits (driven by airflow of 3–6 cm s$^{-1}$ [14]) is simplified by imposing the internal branched HCP, representing the higher capacity of the chimney and lateral channels to transport heat and mass.

The optimum mound configuration is the one that leads to a minimum thermal resistance in the whole body and a minimum nest temperature. To achieve this goal, *all* features of the mound geometry are allowed to freely morph, and it was assumed that the cement (i.e. termite saliva) used in the mounds' construction is strong enough to keep the body together in any circumstances. To investigate the effects of the mound internal channels, several analyses were performed, including analysing a structure with no internal channels, a structure with only a central chimney and a structure with all internal channels shown in figure 1. For the sake of generality, the equations discussed below are provided for the case with all internal channels.

Under steady state and uniform material properties, the dimensionless 2D diffusion equation, shown in equation (3.1), is solved, where $\theta\,(=(T - T_{\mathrm{amb}})/(q_1/(k_0 W)))$ is the normalized temperature ($T$ (K)) field, $k_0$ (W m$^{-1}$ K$^{-1}$) is the thermal conductivity of the body, $W$ (=1 m) is the depth of the body in the third dimension to the plane and $T_{\mathrm{amb}}$ is the ambient temperature. The below equation and its boundary conditions are derived based on those for the dimensional diffusion equation, explained in appendix B

$$\frac{\partial^2 \theta}{\partial \tilde{x}^2} + \frac{\partial^2 \theta}{\partial \tilde{y}^2} = 0. \tag{3.1}$$

The boundary conditions for the above equation are as follows: $-\tilde{k}\partial\theta/\partial\tilde{y} = 1/\tilde{D}_0\cos(\gamma)$ and $-\partial\theta/\partial\tilde{y} = 1/\tilde{D}_0\cos(\gamma)$ at the nest location for the cases with and without internal channels, $\partial\theta/\partial\tilde{n} + \tilde{h}\theta = \check{I}^*$ and $\tilde{k}\partial\theta/\partial\tilde{n} + \tilde{h}\theta = \check{I}^*$ at the mound and HCP surfaces, and $\partial\theta/\partial\tilde{n} = 0$ for the adiabatic surfaces. Here, $\tilde{x}$, $\tilde{y}$ and $\tilde{n}$ are the normalized horizontal, vertical and normal to the surface directions, respectively, $\tilde{D}_0$ is the normalized HCP width of the chimney and '*' denotes the irradiance normal to the face. For the environment-exposed boundary conditions, the heat transfer coefficient, $h$ (W m$^{-2}$ K$^{-1}$), was normalized by the body conductivity ($k_0$) and the square root of the total mound area ($A$), resulting in its dimensionless formulation, $\tilde{h} = hA^{1/2}/k_0$. Likewise, $\check{k}\,(= h_{\mathrm{HCP}}A^{1/2}/k_0)$ is the

normalized thermal conductivity of the HCP that takes into account the convective effects of the air inside the mound, while $\tilde{I} = (I_c A^{1/2} W)/q_1$ is the normalized irradiance. All geometrical parameters with length dimension are normalized using the square root of the whole-body area ($A$), which is a constraint in the analysis, given by

$$A = H' \frac{(L + L'/\cos(\gamma))}{2} - L'^2 \tan(\gamma) + (H - H') \frac{L'}{\cos(\gamma)}$$
$$- \frac{1}{2} \left( \frac{H - H'}{\cos(\gamma)} \right)^2 \tan(\alpha) - \frac{1}{2} \left( \frac{H - H' - L'/\sin(\gamma)}{\cos(\gamma)} \right)^2 \tan(\alpha) \tag{3.2}$$

In its dimensionless form, the area constraint becomes

$$1 = \tilde{H}' \frac{(\tilde{L} + \tilde{L}'/\cos(\gamma))}{2} - \tilde{L}'^2 \tan(\gamma) + (\tilde{H} - \tilde{H}') \frac{\tilde{L}'}{\cos(\gamma)}$$
$$- \frac{1}{2} \left( \frac{\tilde{H} - \tilde{H}'}{\cos(\gamma)} \right)^2 \tan(\alpha) - \frac{1}{2} \left( \frac{\tilde{H} - \tilde{H}' - \tilde{L}'/\sin(\gamma)}{\cos(\gamma)} \right)^2 \tan(\alpha) \tag{3.3}$$

with normalized parameters shown by '~'. The lateral and top surfaces of the body are subjected to convective cooling that is governed by the wind incident on the surfaces. The heat transfer coefficient, $h$ (W m$^{-2}$ K$^{-1}$), is defined based on the ASHRAE [26] correlation, shown in equation (3.4), that takes the effects of surface roughness and wind direction into account. The selected correlation corresponds to a rough surface [26]. $u$ (m s$^{-1}$), denoting the component of the wind velocity ($u_0$ (m s$^{-1}$)) normal to the surface, depending on the incidence angle between the incoming and the surface

$$h = 12.49 + 4.065u + 0.028u^2. \tag{3.4}$$

In situations where the spire completely shades a surface, all direct irradiance is blocked, with only diffuse radiation being perceived by that surface. Alternatively, if there is partial shading over a surface, the amount of direct irradiance over the surface is directly proportional to the portion of the surface that is exposed to the Sun.

In set-ups with the internal channels, the metabolic heat is channelled through the HCP, while in cases without HCP, the metabolic heat is directly diffused in the body from the bottom at a width of $L/3$, in accord with an observation by Turner [18] that indicated the nest size to be roughly one-third of the width of the base. The ratio of the pathway area ($A_p$, equation (3.5)) to the area of the body ($\phi = A_p/A$) introduces a new constraint to the system, shown in equation (3.6)

$$A_p = D_0 H' + D_0 \frac{(H - H')}{\cos(\gamma)} - \frac{1}{2} D_0^2 \sin(\gamma) \cos(\gamma) + D_1(L' - D_0 \cos(\gamma))$$
$$- D_1 \frac{(H - H' - L' \sin(\gamma))}{\cos(\gamma)} \tan(\alpha) \tag{3.5}$$

and

$$\phi = \tilde{D}_0 \tilde{H}' + \tilde{D}_0 \frac{(\tilde{H} - \tilde{H}')}{\cos(\gamma)} - \frac{1}{2} \tilde{D}_0^2 \sin(\gamma) \cos(\gamma) + \tilde{D}_1(\tilde{L}' - \tilde{D}_0 \cos(\gamma))$$
$$- \tilde{D}_1 \frac{(\tilde{H} - \tilde{H}' - \tilde{L}' \sin(\gamma))}{\cos(\gamma)} \tan(\alpha) \tag{3.6}$$

Ultimately, the objective of this analysis is to investigate the optimum configuration of the mound, determined by its geometric parameters ($H$, $L$, $H'$, $L'$, $D_0$, $D_1$, $H_1$, $\alpha$ and $\gamma$), for a given area of the body ($A$), convective heat transfer coefficient ($h$), nominal wind speed ($u_0$), solar irradiance ($I_C$), and solar zenith angle ($\chi$). The optimum configuration is the one that leads to minimum values for the normalized nest temperature ($\theta_{\text{nest}} = (T_{\text{nest}} - T_{\text{amb}})/(q_1/(k_0 W))$). Since in *M. michaelseni* mounds, the over-ground structure is uninhabited and the termite colony resides in the subterranean nest, the minimization of the nest (rather than the whole body) temperature was selected as the objective function for the optimization process. Numerically, the nest temperature quantities were obtained by averaging temperature values in the mesh elements located in the heat-input zone at the base of the structure. Note that $\theta_{\text{nest}}$ can be seen as the thermal resistance of the configuration, given it is directly related to the capacity of the whole system to diffuse the metabolic heat.

The diffusion mechanism is the same for heat and scalar and can be described by the same governing equation. For this reason, the gas-exchange process is modelled in a similar fashion to the heat transfer process, but with different criteria. In this case, the boundary conditions of the surfaces exposed to the environment are of zero scalar concentration with no external sources. The two heat and scalar concentration optimization processes were performed independently. This additional step was made to investigate whether different criteria for the diffusion process in the dense body of termite mound lead to a different conclusion.

## 3.1. Degrees of freedom

Six degrees of freedom ($H/L$, $L'/L$, $D_0/L$, $H_1/H$, $\alpha$ and $\gamma$), representing the structural characteristics of the mound, are chosen. $H/L$ defines the height-to-base width ratio of the mound. $L'/L$ indicates the width of the spire to the width of the base. With these two, aligned with $\gamma$ and $\alpha$, it is possible to define the external mound structure. $D_0/L$ relates the width of the internal pathway to that of the base, and $H_1/H$ defines the branching height of the lateral channels to the total height of the mound. For a given value of $\phi$ (=0.18 for all the analyses) and these degrees of freedom, the system can be fully defined.

## 3.2. Numerical procedures

Equation (3.1) is solved by applying a finite-element method with non-uniform, triangular elements, implemented in MATLAB environment, precisely the partial-differential equations (PDE) toolbox [27]. An appropriate mesh size was determined using successive refinements, in which the number of elements was increased four times until the criterion $|(\theta_{\max}{}^j - \theta_{\max}{}^{j+1})/\theta_{\max}{}^j| < 1 \times 10^{-4}$ was satisfied. $\theta_{\max}^j$ and $\theta_{\max}^{j+1}$ represent the maximum normalized temperatures calculated using the current and the next mesh sizes, respectively. This criterion was satisfied for a mesh composed of approximately 35 000 triangular elements.

For the optimization, a binary, single-objective, elitist genetic algorithm (GA) was used, similar to previous work in the literature [28,29]. The algorithm creates a fixed number (population) of geometries, which are generated randomly, called the first generation (set of individuals). Then, the algorithm calculates the performance of each geometry concerning the minimization of nest temperature. After that, a portion of the geometries with the lowest net temperatures are selected (selection) to generate the population of the next generation (crossover). In the next step, the algorithm has a possibility to update values of the free parameters of individuals from this new population, except in the elite individuals (mutation). Finally, the algorithm checks the stopping criteria, and, if they are not met, creates a new generation, repeating the process. In this work, the algorithm used has a population type consisting of a bitstring (binary GA), with populations composed of 60 individuals. The selection was uniform, with two elite individuals. Eighty per cent of the new population is created by crossover, while the other 20% are created via mutations. There is a 10% chance for a mutation in each degree of freedom of the mutated individuals. The stopping criterion is based on the number of generations without improvement in the performance of the best individual. When the algorithm goes over 20 subsequent generations without improvement, it stops. The algorithm chart is included in appendix C.

Results discussed in the following sections are based on the best individual, i.e. the best geometry that arises from the GA optimization. The range and resolution of each degree of freedom are presented in table 1 ($H'$ and $D_1$ are not included, given that they are found by the total area and the pathway area constraints). To account for the uncertainties within the resolution of each degree of freedom, the values found in the following chapters are plotted with error bars, whose height is determined by the resolutions indicated in table 1.

# 4. Model validation for different structural set-ups

The typical environmental condition of Outjo in Northern Namibia (16°4.50′ E, 19°59.05′ S) was chosen as the base environmental condition for the *M. michaelseni* termite mounds, exhibiting a yearly average solar irradiance ($I_c$) of 700 W m$^{-2}$ [30], with yearly average zenith angle ($\chi$) of 19° at noon, and yearly average incoming wind speed ($u_0$) of 1.38 m s$^{-1}$, blowing from the south [5] (left side of the mound model). The typical metabolic heat ($q_1$) for termite mounds is considered 50 W [23], representing a mature mound, which results in a normalized irradiance ($\tilde{I}$) of 14.

**Table 1.** Resolution and range for each degree of freedom used in the optimization process.

| degree of freedom | resolution | range |
|---|---|---|
| $\gamma$ | 1 | $-45°$–$45°$ |
| $\alpha$ | 1 | $0°$–$30°$ |
| $D_0/L$ | 0.005 | 0.005–0.5 |
| $H_1/H$ | 0.05 | 0.05–0.95 |
| $L'/L$ | 0.05 | 0.05–0.95 |
| $H/L$ | 0.1 | 0.1–10.0 |

To investigate the performance of the model, the optimum architecture of the mounds with and without the internal channels was found for the base environmental conditions and compared against the architectural characteristics of a typical *M. michaelseni* mound, reported in the literature. While quantitative data of geometrical characteristics of termite mounds are scarce, in general, Turner [18] provided measured data on the average height, base width, mid-height width and spire tilt angle of *M. michaelseni* mounds. These measured data in the dimensionless form defined in this work yield $\tilde{L} = 1.5$, $H/L = 0.9$, $L'/L = 0.423$ and $\gamma = 19°$. These values were used as reference data in this study. Based on this information, the dimensionless base width ($\tilde{L}$) was fixed to 1.5 in the analyses. Additionally, for all numerical analyses performed, the optimum configuration was taken as the one with the best performance (i.e. yielding minimum nest temperature), after the GA was run three times, independently.

## 4.1. Mound with no internal channels

With the base environmental conditions, the optimum mound architecture for the case with no internal channels resulted in the height-to-base ratio ($H/L$) of 1.5 (+67% deviation from the reference data), spire width-to-base ratio ($L'/L$) of 0.25 (−40% deviation from the reference data), funnelling angle ($\alpha$) of 0° ($\alpha$ is visually greater than zero, but no specific data are reported in the literature) and inclination angle ($\gamma$) of 27° (+42% deviation from the reference data), with a normalized nest temperature ($\theta_{nest}$) of 0.49. The temperature distribution over the optimal mound configuration is shown in figure 3.

To minimize the nest temperature, the structure of the mound presents a large aspect ratio, distancing the radiatively heated top surface from the nest location and forming a narrow spire that is convectively cooled by the wind. The slim cooled spire allows for a more effective dissipation of the nest output heat. Due to the Sun's zenith angle fixed at 19° (northwise), the right side of the base receives more direct radiation than the left side. Thus, the long spire tilts to the right to shade the right side of the base. Under this condition, the right side only receives diffuse radiation. The results indicate that the architectural characteristics of the optimum mound with no internal channels significantly differ from the typical characteristics of real mounds reported in the literature.

## 4.2. Mound with internal channels

It is widely agreed that the internal channels of termite mounds play an important role in their thermoregulation process [1,5,7,31,32]. Therefore, in order to better understand the effects of the internal channels, an internal pathway with a heat conductivity higher than that of the body was added to the system. The area ratio of the HCP and the body ($\phi$) was set to 0.18, which was chosen based on the typical volume portion of tunnels in termite mounds, as reported in Turner [18]. The nest output heat is then channelled through the HCP and diffused through the mound body. For the HCP, the heat transfer coefficient for air in free convection, $h_{HCP} = 5 \text{ W m}^{-2}\text{ K}^{-1}$ [33], was considered, and the conductivity of the mound clay was set to $k_0 = 0.2 \text{ W m}^{-1}\text{ K}^{-1}$ [25]. Therefore, the dimensionless thermal conductivity within the channels, $\tilde{k}(= h_{HCP}A^{1/2}/k_0)$, was determined to be 25.

The optimal mound structure obtained for the base environmental condition exhibits features of $H/L = 1$ (+11% deviation from the reference data), $L'/L = 0.45$ (+6% deviation from the reference data), $\gamma = 12°$ (−37% deviation from the reference data), $\alpha = 0°$ and $D_0/L = 0.06$ (reference data are not reported in the literature), with a minimized dimensionless nest temperature ($\theta_{nest}$) of 0.21 (57% less than that in the mound with no internal channels). The mound structure and its temperature distribution are shown in figure 4. A strong

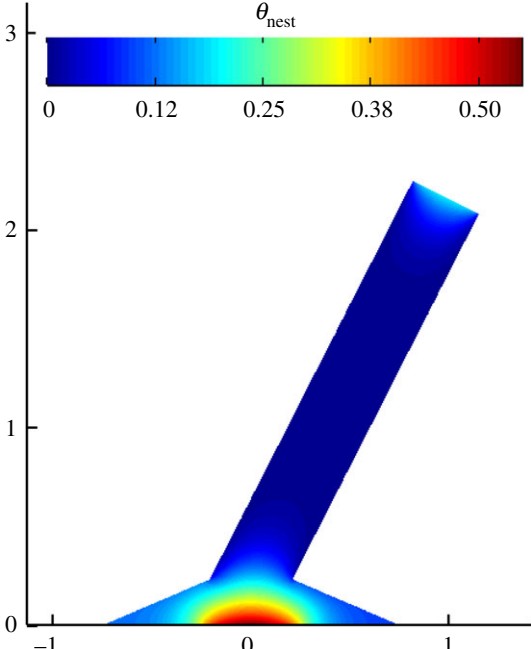

**Figure 3.** Dimensionless temperature distribution over the optimal mound architecture with no internal channels, achieved for the typical environmental conditions of Outjo, northern Namibia (direct solar irradiance of 700 W m$^{-2}$, wind speed of 1.38 m s$^{-1}$ and solar zenith angle of 19°), and nest output heat rate of 50 W.

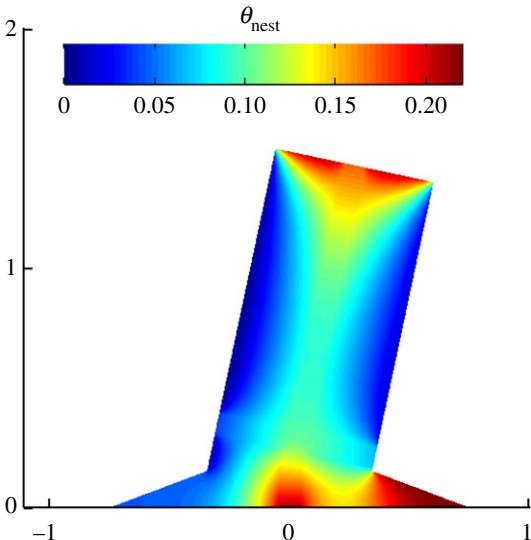

**Figure 4.** Dimensionless temperature distribution over the optimal mound architecture with internal channels, achieved for the typical environmental conditions of Outjo, Northern Namibia (direct solar irradiance of 700 W m$^{-2}$, wind speed of 1.38 m s$^{-1}$ and solar zenith angle of 19°), and nest output heat rate of 50 W.

agreement can be seen between the dimensions of a typical mound and the modelled configuration, with the error (the difference between the values found and those of the reference data) of the optimized structure being within the resolution of the optimization method (table 1). Even though the mound inclines to the north, the inclination angle deviates from the observations of Turner [18], who found the *M. michaelseni* termite mound average inclination to be 19°—equal to the local yearly average solar zenith angle. This deviation can, partially, be attributed to the value considered for the dimensionless irradiance (the ratio of solar irradiance and the nest input heat) for the base environmental condition (this is further discussed in §5). In addition, the optimal configuration here does not show funnelling in the spire. Overall, the resulting configuration indicates the good performance of the model and its capability in replicating the termite mound configuration under the most relevant thermal forces.

An optimization analysis for gas concentration in the mound was also performed, with the optimal mound being the one that has the lowest scalar concentration in the nest. The resulting optimal configuration (not shown) from this analysis was similar to those found from the thermal analysis, with the main difference between the two being in the mound spire inclination. For gas concentration, the spire was vertically oriented ($\gamma = 0°$), indicating that the mound spire inclination arises due to the thermal processes.

The addition of internal channels significantly alters the mound structure and the temperature regulation process within it. With this adjustment, the mound structure shortens in height and widens in the spire, providing a more suitable architecture to balance the metabolic and external heats, therefore regulating the nest temperature more effectively. In this configuration, the nest heat is directed mostly through the channels, leading to a significant reduction in the nest temperature (compared to that in the mound with no internal channels). The reduction in the height of the mound is related to the size of the central chimney, which, given the area constraint, decreases the height of the structure to become wider without narrowing the lateral channels as much.

Another feature of the optimal mound configuration is the presence of lateral channels in the spire, not in the base. However, the lateral channels tend not to appear at the top of the spire, but around the centre of the mound structure. While this might seem contradictory, given that Turner [18] observed the proportion of the channels at the top of the spire is considerably higher than that at the spire bottom and in the base, this higher proportion occurs due to the spire being narrower at the top, while in our model, the spire width remained constant (funnelling angle, $\alpha = 0°$) throughout its extension. As a counterpoint, Turner [18] also observed that the central region of the mound is where the branching of the lateral channels occurs more prominently (the same feature that is noted, here).

From a heat transfer point of view, having lateral channels closer to the nest allows for both metabolic and solar heats to be conducted more effectively away from the nest location. Yet, if the lateral channels were in the base, they would have to extend longer, laterally, making the central chimney slim (due to the area-ratio constraint) and, thus, ineffective in removing the metabolic heat. Moreover, being in the spire allows the lateral channels to harness the wind energy more effectively and cool the south (windward) face of the mound. In turn, this also influences the inclination of the spire, allowing it to become tilted to the north and shade the leeward side of the base, therefore decreasing the intensity of the incoming irradiance over it. The spire does not show any funnelling, as this would increase the overall height of the mound, due to the area constraint. Therefore, the width of the lateral channels would decrease, which would hinder the diffusion of the heat from the top surface. The implication of funnelling is further discussed in §5.1.5.

To investigate the effect of the lateral channels on the mound configuration, the study was extended to a set-up with only a central chimney. The optimal structure obtained for the base environmental condition exhibits a configuration with $H/L = 0.8$ (−11% deviation from the reference data), $L'/L = 0.55$ (+30% deviation from the reference data), $\gamma = 3°$ (−84% deviation from the reference data), and $\alpha = 0°$, with a minimum dimensionless nest temperature ($\theta_{nest}$) of 1.003. In this configuration, while the optimized mound shape significantly deviates from that of a typical mound, a significant increase in nest temperature is also noted (378% greater than that in the mound with both the chimney and internal channels). This increase in nest temperature occurs because without the lateral channels, the only effective path to diffuse the metabolic heat is the central chimney, which opens to the top of the spire, where the solar heat channels through. Due to the existence of only one main diffusion channel, the optimal mound configuration is the one that balances the distance between the nest and the top surface.

The results indicate that the mound model with all internal channels (central chimney and lateral channels) predicts the structural features of a typical *M. michaelseni* mound, when the optimization process is performed with the typical environmental values of these termites' habitat. To explain the mounds' geometrical features and behaviours observed in nature, the study was extended to different environmental conditions, namely, different solar irradiances, zenith angles and wind intensities, as explained in the following section.

# 5. Results and discussion

The optimal configuration for the mound with no internal channels was found to show negligible responses to the environmental conditions. Therefore, only the results for the mound with internal channels are reported below.

## 5.1. Influence of environmental conditions on the mound architecture and thermal performance

### 5.1.1. Solar irradiance

To study the effect of solar irradiance on the mound geometry and thermal performance, the irradiance ratio ($\tilde{I}$) is changed between 0 and 150. This range accounts for cases with no or small ($I_c = 0 - 175$ W m$^{-2}$), mild ($I_c = 350$ W m$^{-2}$) and strong ($I_c = 700$ W m$^{-2}$) solar irradiances typical at the region of northern Namibia [30] throughout the year, and for mature and small mounds that produce 50 and 4.7 W of metabolic heat [23]. Values of wind speed and zenith angle were fixed to the typical values of Outjo, Namibia ($u_0 = 1.38$ (m s$^{-1}$), $\chi = 19°$).

The results indicate that the spire inclination angle ($\gamma$) is strongly influenced by the amount of solar irradiance. Under the extreme solar irradiance ratio ($\tilde{I} = 150$), the mound architecture tends to be taller and has a pronounced spire inclination, with a value equal to the imposed solar zenith angle, while mounds located in less exposed regions (experiencing weaker solar irradiance) are shorter and more vertically oriented. As shown in figure 5, for $\tilde{I} = 0 - 150$, the spire inclination angle ranges between $\gamma = 1°$ and $\gamma = 19°$. This finding is strongly in agreement with Turner's observation, where he reports that mounds located under tree shadows are more upright, while those exposed to the Sun exhibit a spire tilt of 19° (same angle as the local latitude) [18,34]. In open areas, mounds become more sensible to solar heat, especially over the base surface on the right (north) side (note that in the Southern Hemisphere, northward surfaces receive more direct irradiance). Hence, these mounds respond in a way to shade the north base surface, while also distancing the nest from the external heat transferred through the central chimney. This manifests through a more prominent spire inclination.

In addition, the results indicate that being exposed to higher solar irradiances leads to the presence of wider lateral channels, representing a more complex internal structure and, in turn, a narrower central chimney (figure 6), as they are related in the model owing to the area constraint. In these regions, wider lateral channels provide a much more effective way to disperse the heat from the mound surfaces. While the central chimney experiences a high-intensity solar irradiance coming from the top, narrowing the central channel creates more resistance for the heat, making a difficult path for it to reach the nest. By consequence, the heat is directed to the wider lateral channels to be dissipated through convective cooling.

### 5.1.2. Solar zenith angle (the effect of geographical location)

The spires of *M. michaelseni* mounds located 22 km north of Outjo, in northern Namibia (16°4.50′ E, 19°59.05′ S) have a northward tilt angle of 19°, on average [18]. The similarity between the spire tilt angle and the geographical latitude led to the hypothesis that the Sun's position in the sky might be a determinant factor for the spire tilt [34]. This premise is also noted in the study of Darlington [4], who observed that the mounds in Kajiado, Kenya (36°48′ E, 1°50′ S) were only slightly tilted to the west (away from the prevailing wind), but not to the north. To investigate this effect, different zenith angles ($\chi$) ranging between 0° and 25° are analysed with fixed wind speed ($u_0 = 1.38$ m s$^{-1}$) and irradiance ratio ($\tilde{I} = 14$). The irradiance ratio of 14 represents a yearly average irradiance of 700 W m$^{-2}$ for a mound of 50 W.

Results indicate that the inclination of the spire decreases with decreasing zenith angle, showing that mounds closer to the equator ($\chi = 0°$) are almost upright ($\gamma = 2°$), while for $\chi \geq 15°$, mounds tend to have a more prominent spire inclination ($\gamma = 12°$). These results support the theory proposed by Turner [34]. To strengthen this claim, the tilt angle behaviour was investigated further by analysing different zenith angles for the extreme value of irradiance ($\tilde{I} = 150$). Results showed that the tilt angle of the spire *consistently* follows the amount of zenith angle. Thermally, the inclination in the spire allows the structure to partially shade the northward face of the mound and block a large portion of the direct solar radiation over its surfaces. This allows the nest temperature to be regulated for different geographical locations, as the dimensionless temperature changed 7% within the zenith angle range analysed. The effect of solar zenith angle over the spire inclination is shown in figure 7.

Mounds located closer to the equator line receive more prominent heat at their top surface and, therefore, they tend to be taller and, consequently, narrower in the spire ($H/L = 1.3$, $L'/L = 0.35$ for $\chi = 2$) to create more resistance for the heat to reach the nest. On the other hand, mounds that experience solar heat more prominently on the lateral surfaces are prone to present a compact architecture with a wider spire ($H/L = 1$, $L'/L = 0.45$ for $\chi = 12$). This arrangement distances the

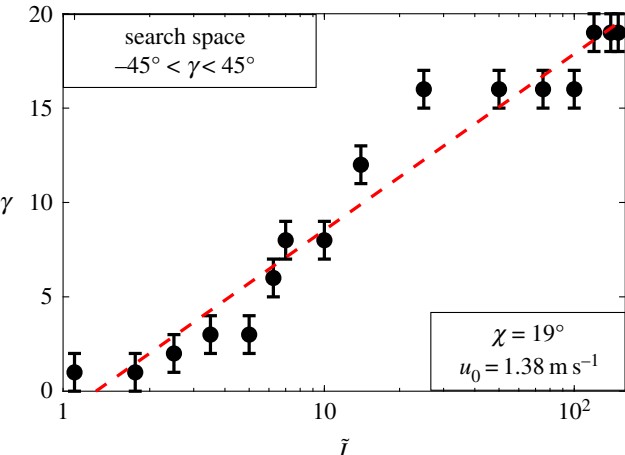

**Figure 5.** Effect of the solar irradiance ratio ($\tilde{I}$) on the spire inclination ($\gamma$) for a wind speed of $u_0 = 1.38$ m s$^{-1}$ and zenith angle of $\chi = 19°$. The red line and the error bars, respectively, represent the overall trend and the numerical uncertainty of the results within the resolution of the optimization method.

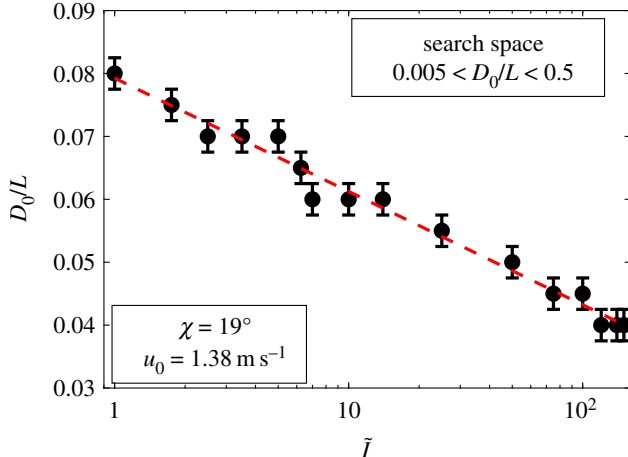

**Figure 6.** Effect of the solar irradiance ratio ($\tilde{I}$) on the width of the central chimney ($D_0/L$), for a wind speed of $u_0 = 1.38$ m s$^{-1}$ and zenith angle of $\chi = 19°$. The red line and the error bar, respectively, represent the overall trend and the numerical uncertainty of the results within the resolution of the optimization method.

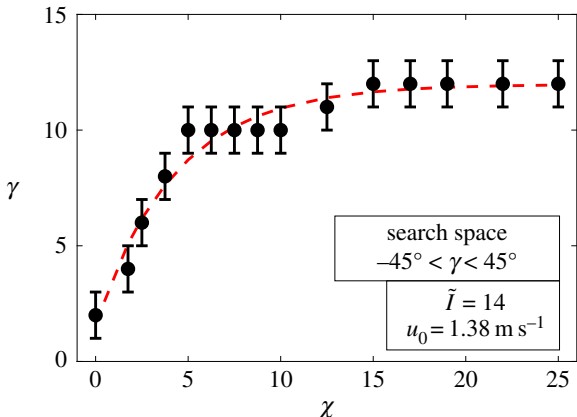

**Figure 7.** Influence of solar zenith angle ($\chi$) over the optimal inclination of the mound spire ($\gamma$) for irradiance ratio $\tilde{I} = 14$ and wind speed $u_0 = 1.38$ (m s$^{-1}$). The red line and the error bar, respectively, represent the overall trend and the numerical uncertainty of the results within the resolution of the optimization method.

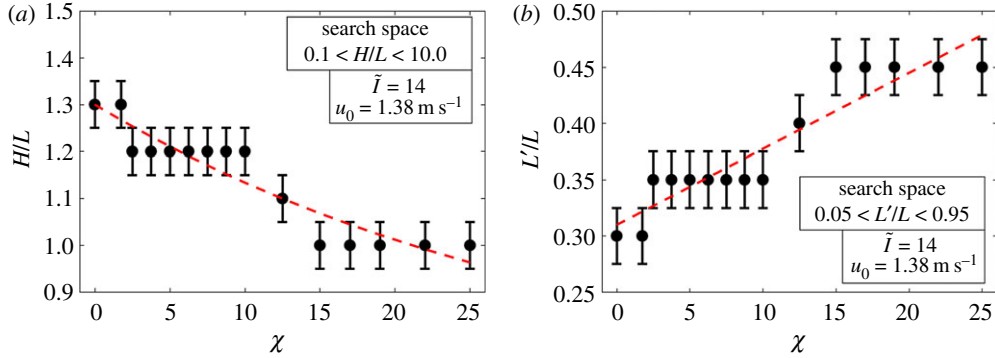

**Figure 8.** Effect of solar zenith angle on (*a*) the optimal mound height, represented by $H/L$, and (*b*) spire width, represented by $L'/L$. The red line and the error bar, respectively, represent the overall trend and the numerical uncertainty of the results within the resolution of the optimization method.

lateral surfaces from the nest and, given that these surfaces experience a higher solar irradiance, allows the nest temperature to be more effectively regulated. The effect of different solar zenith angles on the mound shape is shown in figure 8.

### 5.1.3. Wind

Wind energy is known to be one of the driving forces for the nest ventilation (e.g. [5]). To understand the wind effect on the mound geometry, different wind speeds are considered, ranging from 0 to 5 m s$^{-1}$ [5], while assuming fixed values for $\tilde{I}$ (= 14) and $\chi$ (= 19°). For these analyses, logarithmic and constant wind velocity profiles were investigated, with results showing no noteable difference between them (not shown). Therefore, the results are presented for the constant wind profile.

For the range of values investigated, the results indicated that the optimal mound structure remained the same for different wind speeds, indicating that the mound is capable of convectively diffusing the heat from the mound surfaces, without a need for altering the overall mound architecture. Although wind does not modify the external structure of the mound, it has a significant influence on the size of the internal channels. Mounds that experience stronger winds tend to have more capacious lateral channels, while also presenting a reduced central chimney. This increase in the lateral channels allows them to better harness the wind energy. At the same time, mounds located in regions with no or weak wind speeds display a larger central chimney to allow the metabolic heat to be diffused through a larger channel area (figure 9).

Besides its effect on the internal channels, wind can play a significant role in the nest temperature. Strong winds around the mound can cool down the mound body and, by consequence, the nest. The nest temperature of a mound that experiences a wind of $u_0 = 5$ (m s$^{-1}$) is around 14% lower than that of a mound exposed to no wind. This reduction in temperature might be an adverse effect for the termites, as they reverse this temperature reduction by building their mound in a more compact, dome-like shape [16]. Such a compact configuration capitalizes on the metabolic heat from the nest to increase its temperature and, therefore, it has a different objective function that is not considered in this work. Figure 10 summarizes the optimal mound configurations for different environmental conditions explained above.

It is noted that in all cases analysed above, the funnelling angle of the optimized configurations remained unaffected. Additional simulations were set up to investigate whether the funnelled spire of termite mounds have any contribution to the thermal processes within the mound, and whether the lateral channels that contribute to heat dissipation midway between the nest and the top surface affect this result.

### 5.1.4. Effect of the lateral channels

A new set of simulations were performed for a mound with only a central chimney to investigate the effect of the removal of the lateral channels on the mound geometry. It is found that the mound optimal configuration remained unaffected to the wind and zenith angle parameters, while significantly influenced by the amount of the irradiance ratio. In contrast with the mound with chimney and lateral channels, the mound without lateral channels is much more sensitive to the intensity of the external heat. That is because, in this case, the chimney is the main passageway to balance the effect of the two

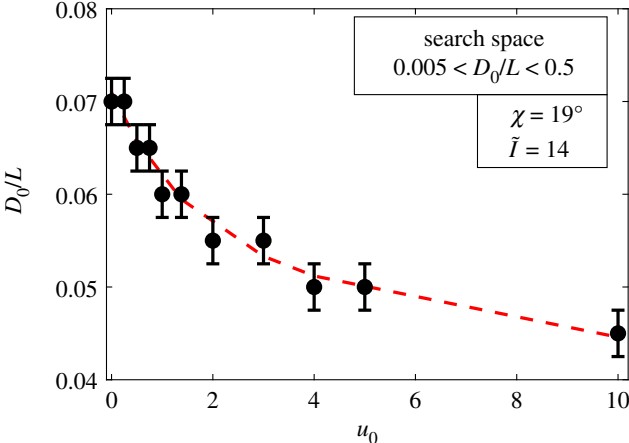

**Figure 9.** Effect of wind speed on the width of the mound central chimney, represented by $D_0/L$. The red line and the error bar, respectively, represent the overall trend and the numerical uncertainty of the results within the resolution of the optimization method.

heat sources (solar and metabolic), and removal of the lateral channels removes additional effective paths for these heats. As a result, unlike the case with lateral channels, the mound geometry revealed a significant sensitivity to the amount of the irradiance, reflected in the quantities of mound aspect ratio, spire tilt and funnelling angles. The mound aspect ratio varied between 0.6 (for $\tilde{I} = 1$) to 1.1 (for $\tilde{I} > 50$), and the spire tilt angle changed from $\gamma = 1°$ ($\tilde{I} = 1$) to $\gamma = 31°$ ($\tilde{I} > 75$). These large variations appear because, under stronger solar irradiances, taller mounds create greater distances, thus larger resistances, between the nest and the top surface. For higher irradiance ratios, the external heat overwhelms the metabolic heat and becomes more important in increasing the mound temperature. Thus, to lessen the effects of this heat, the mound distances the top surface by becoming taller and more tilted, as there are no lateral channels to effectively diffuse the external heat.

In contrast with the previous configurations, the mound without lateral channels exhibited funnelling in the spire for $\tilde{I} > 50$, increasing until it stabilized at $\alpha = 12°$ for $\tilde{I} > 100$. The reason behind the appearance of funnelling angle, in this case, is the mound's need for becoming taller under strong solar heats. By increasing the funnelling angle, the mound uses the last resource of its constrained body to become taller and separate the top surface further away from the nest. In contrast with the tilting angle, this effect manifests only at high irradiance ratios, indicating that it is more thermally efficient for the mound to tilt its spire than funnel it.

In addition to studying the effects of the removal of the lateral channels, the influence of having several lateral channels with independent heights was also investigated. The inclusion of several lateral channels did not affect the mound architecture and altered the nest temperature compared to those with one set of channels by only $\Delta\theta = 0.01$. Considering the small sensitivity of the model to this additional complexity and the amount of additional computational efforts, mounds with more than one set of lateral channels were not analysed further.

### 5.1.5. Effect of forcing funnelled spire on the thermal process of the mound

To further investigate the absence of spire funnelling in the mound optimum configurations, the area constraint was relaxed for the mound with lateral channels and all the aspects of the optimal configuration (§4.2), except the funnelling angle, were kept unchanged. The results of these investigations indicated that the nest temperature did not alter ($\Delta\theta = 10^{-4}$) in response to the forced funnelling. This shows that the configurations previously obtained are not just thermally optimized, but they also have a certain resilience in maintaining their thermal performance. It could be concluded that the spire funnelling may not arise due to thermal processes, but rather due to different mechanisms, such as mechanical stability and erosion (not considered in this work).

## 5.2. Influence of porosity on the mound architecture and thermal performance

The weak convective process in the mound conduits is driven by an airflow of 0.03–0.06 m s$^{-1}$ [14], while the rest of the mound body is impervious to bulk flow. The mound body is porous on a microscopic scale

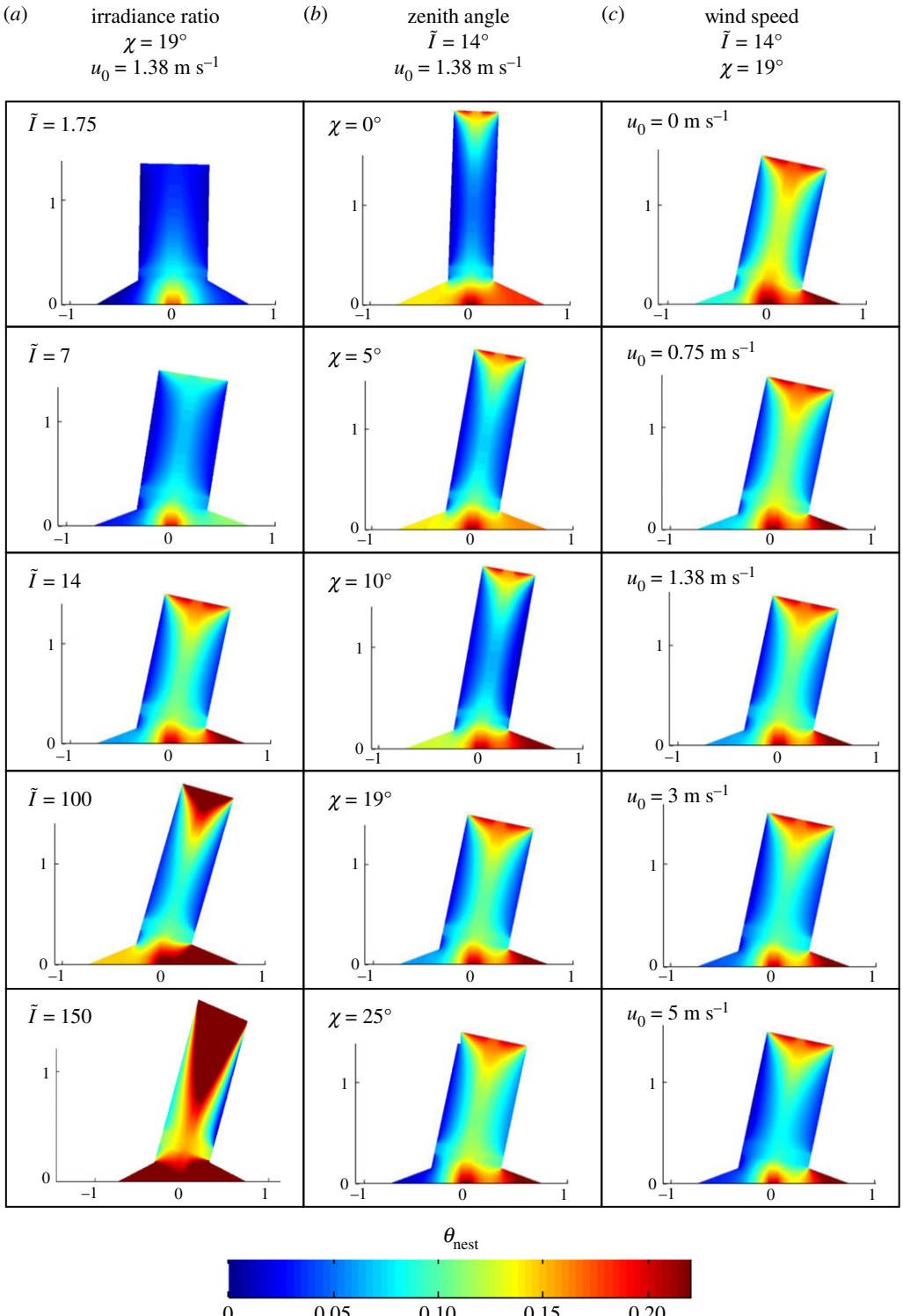

**Figure 10.** Dimensionless temperature distribution of the optimal termite mound configurations for different solar irradiances (*a*), solar zenith angles (*b*) and wind speeds (*c*) (the same colour bar was used for all configurations).

that only allows for diffusive exchanges (personal communication with Dr Hunter King, 2019). This evidence led the study to consider the mound model to be composed of a solid body with a conductivity of clay and internal channels with higher conductivity to represent the higher capacity of the conduits to transfer heat and mass (see §3). However, to investigate the effect of the porosity of the mound soil-based body on its thermal performance and architecture, the study was extended as described below.

The effect of porosity is considered by using porosity-based effective thermal conductivity ($\tilde{k}_{\mathrm{eff}}(\rho)$). The effective thermal conductivity is calculated based on a linear interpolation between the thermal

conductivity of the soil and the air, and in its dimensionless form, it is defined as $\tilde{k}_{\text{eff}}(\rho) = 1 + (\tilde{k} - 1)\rho$, where $\rho$ is the local porosity and $\tilde{k}$ is the normalized thermal conductivity of the soil. The performance of the model was, initially, tested for the typical environmental conditions in Outjo, Namibia, for a mound with half of the total porosity (e.g. 18%) distributed in the body and the other half concentrated in the internal channels. The optimum configuration obtained in this case (i.e. $H/L = 1$, $L'/L = 0.45$, $D_0/L = 0.04$, $\gamma = 12°$, $\alpha = 0°$, with a minimized dimensionless nest temperature ($\theta_{\text{nest}}$) of 0.21) was similar to that found for the mound with a solid body (§4.2). The investigation was extended for different values of environmental forces. It was found that the optimum mound structure remained insensitive to different values of wind speed and zenith angle. However, the spire tilt increased significantly ($\gamma = 29°$) for higher values of irradiance ratios ($\tilde{I} \geq 50$), deviating from those of the solid mound ($12° < \gamma < 19°$ for $\tilde{I} \geq 100$). This change in the behaviour of spire tilt is due to the lower effective conductivity of the mound body in the porous case. The lower conductivity results in a concentration of heat around the nest area, especially on the north side, where the Sun heats the base. Therefore, the mound becomes more inclined to better shade the north face of the base.

# 6. Conclusion

In this work, a computational model has been developed to predict the effect of environmental conditions on the form and function of a natural structure—termite mound. The proposed model considers the most relevant forces involved in the heat transfer process of termite mounds and adopts a system configuration procedure to determine an optimized mound structure. Due to the availability of the information, the study used the structural characteristics of the mounds of *M. michaelseni* termites. However, the proposed methodology can be applied to any natural structure, where thermodynamics and heat transfer play an important role in their functions.

A series of simulations were conducted to confirm the accuracy of the model under the typical environmental conditions of *M. michaelseni* termite habitat in northern Namibia. The optimal configuration that arose from the model was found to be strikingly similar to those observed in nature, including the mound spire being inclined towards the Sun—as noted in Turner's observation [34]. The study was then extended to examine the mound configurations under different values of wind, solar irradiance and zenith angle (representative of geographical latitude). The results indicate that the mound superstructure and its internal condition strongly depend on the combined effect of all environmental forces. In general, different amounts of exposure to direct insolation lead to dissimilar internal and external mound features. Mounds being exposed to higher solar irradiances develop wider lateral channels, inside, and taller and more pronounced spire tilt, outside. Moreover, for extremely high solar irradiances, the mound points exactly to the average zenith angle, indicating that the spire inclination is a consequence of thermal processes within the mound. This is in agreement with the hypothesis proposed by Turner [34] and with reported field observations. In nature, *M. michaelseni* mounds located in northern Namibia (19°S) tend to have a prominent spire tilt of 19.6° [18], while mounds studied in Kajiado, Kenya (1°S), do not show any significant tilt northwise [4]. Additionally, we determined that mounds located closer to the equator line tend to be taller and narrower in their upper structure, which matches field observations. For example, *Macrotermes jeaneli* mounds in southern Ethiopia (3°N) [35], reach, in average, 2–5 m in height, presenting a distinctive upright, narrow spire and a large base. By contrast, mounds that are located further away from the equator line were found to have more compact configurations, with shorter and wider structures. For example, this is the case for the *M. michaelseni* in northern Namibia (19°S) [18] (around 2 m high), as well as the *Conitermes cumulans* and *Syntermes dirus* termites from central Brazil (18°S) (around 1 m high) [36].

From our investigation, it was found that while wind does not influence the overall over-ground mound shape, it affects the mound internal structure and temperature. Increasing wind speeds led to wider lateral channels and narrower central chimney, which reduced the nest temperature significantly. In nature, mounds under unfavourable environmental conditions, such as strong winds, can be found with a nest temperature below a comfortable range [16]. To reverse this effect, termites reshape their mound to a more compact shape, with less internal channels, to increase the nest temperature [16]. However, this was not observed in this study since the reshaping process occurred due to a different objective function.

The configuration that arose from the gas-exchange analysis was like those obtained from the thermal study, except in the spire alignment, which was found to be oriented vertically. The similarity between configurations shows that the scalar diffusion did not affect the mound configuration, while the contrast

in spire behaviour provided substantial evidence of the mound inclination being a consequence of thermal processes within the mound, rather than gas exchange.

The effect of the porosity of the mound body was also investigated. It was found that a porous body mound led to configurations analogous to those of a solid body mound. These similarities in the architectures show that the consideration of a solid body with air-filled internal channels is adequate for the analyses presented in this work.

While optimal mound configurations did not exhibit a funnelled spire, further investigations showed that the funnelling in the spire did not have a significant effect on the heat exchange process of the mound. This observation suggests that the presence of this feature in mounds might not be due to thermal processes.

Overall, the mound structure and its internal channels are arranged in such a way as to promote homeostasis of the mound atmosphere [5]. This is seen throughout the different analyses of this work, where each environmental factor displayed a specific effect on the mound structure. From a thermal point of view, the optimal configuration of the mound is the one that facilitates access to the currents that flow through the system, such as the metabolic and solar heats.

While the current research has a focus on the shape and function of termite mounds under relevant thermal forces, it is important to note the importance of other factors, including erosion due to water dynamics and material stability. In addition, mounds play much more intricate roles in creating suitable microclimates for the termites, for example, through water retention [37–39]. The effects of these important factors are left for future studies.

While the focus of the current research has been on one particular species, the methodology could be applied to predict the effect of environmental forces on any natural structure for which structural and environmental information can be obtained. As such, this practice provides a broader view of the factors that are effective in the form and function of a naturally made structure.

Data accessibility. Codes and materials supporting this article have been uploaded to the Dryad Digital Repository: https://dx.doi.org/10.5061/dryad.b75g05b [40].
Authors' contributions. T.M.F. designed the study, conducted the simulations, analysed the results and wrote the manuscript. J.C.O. designed the study, analysed the results and critically revised the manuscript. N.Y. conceived of the study, designed the study, coordinated the study, analysed the results, wrote the manuscript and critically revised the manuscript. All authors gave final approval for publication and agree to be held accountable for the work performed therein.
Competing interests. We declare we have no competing interests.
Funding. This study was supported by funding from the Florida State University and FAMU-FSU College of Engineering.
Acknowledgements. We thank Dr Hunter King for the valuable inputs into the model set-up.

# Appendix A. Justification for the utilization of a 2D model

It was initially assumed that the effect of the external (solar irradiance and wind) and internal (metabolic heat) forces in the third dimension are not vital in capturing the function of the mound. This assumption was mainly due to the observed northward tilt of the mound and its conical shape in nature. In addition, observational studies report that the prevailing wind in the natural location of these mounds is from the south and the dominant solar irradiance is from the north. This assumption was tested by comparing the results from a 2D model against a 3D one.

The two simulations were performed using the same geometrical features and boundary conditions. A comparison between the temperature distribution of the 2D model and a cross-section from the centre of the 3D model is shown in figure 11. The results indicate a strong similarity in the temperature distribution and architecture of the mound between the two models. This similarity indicates that a 2D model can replicate what a 3D model captures, while significantly reducing the computational costs.

# Appendix B. Dimensional diffusion equation and boundary conditions

In the model used in this work, heat is transferred through diffusion in a homogeneous body under steady-state conditions. The governing diffusion equation is

$$\frac{\partial^2 T}{\partial x^2} + \frac{\partial^2 T}{\partial y^2} = 0,$$

where $T$ is the temperature (K), and $x$ and $y$ are the horizontal and vertical directions, respectively.

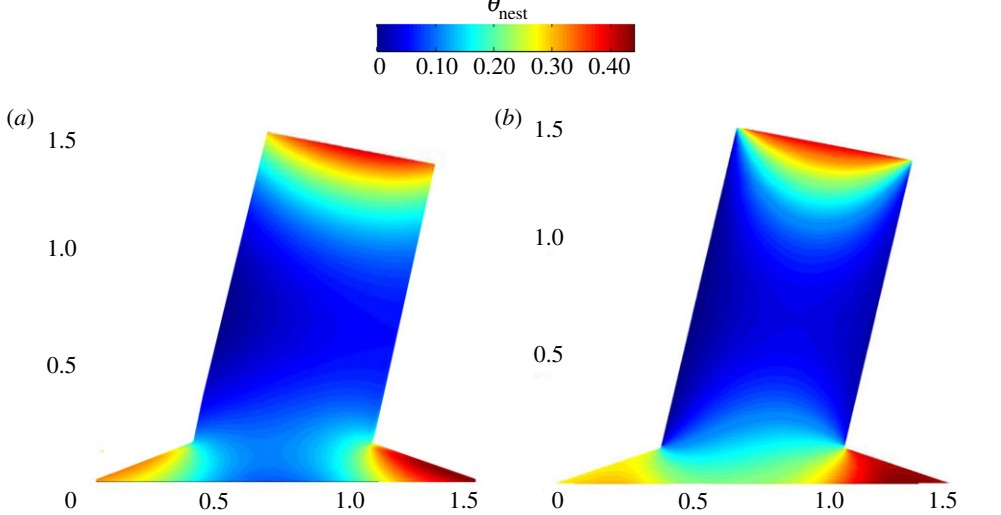

**Figure 11.** A comparison between the temperature distribution of the 2D model (*b*) and a cross-section from the centre of a 3D model (*a*).

The boundary condition at the nest location ($-D_0/2\cos(\gamma) \leq x \leq D_0/2\cos(\gamma)$ and $y = 0$) is a heat rate input defined as $-k_0(D_0 W)(\partial T/\partial y) = q_1$ for the model without internal channels. For the model with internal channels, the heat rate input is located directly at the base of HCP, defined as $-h_{\text{HCP}}A^{1/2}(D_0 W)(\partial T/\partial y) = q_1$, where $h_{\text{HCP}}A^{1/2}$ represents the effective conductivity of the pathway, accounting for the convective effect of air in the channels.

For the external surfaces of the mound, the boundary condition is composed of a balance between convective cooling by the ambient air, incoming irradiance from the Sun and conduction in the mound body. The resulting equation for the boundary condition for the mound surfaces is $k_0 A(\partial T/\partial n) + hA(T - T_{\text{amb}}) - I_c A = 0$, while the boundary condition for the HCP surfaces is $h_{\text{HCP}}A^{1/2}A(\partial T/\partial n) + hA(T - T_{\text{amb}}) - I_c A = 0$. Finally, for the adiabatic surfaces, the dimensional boundary condition is given by $(\partial T/\partial n) = 0$, where $n$ is the direction normal to the surface.

# Appendix C. Genetic algorithm chart

See figure 12.

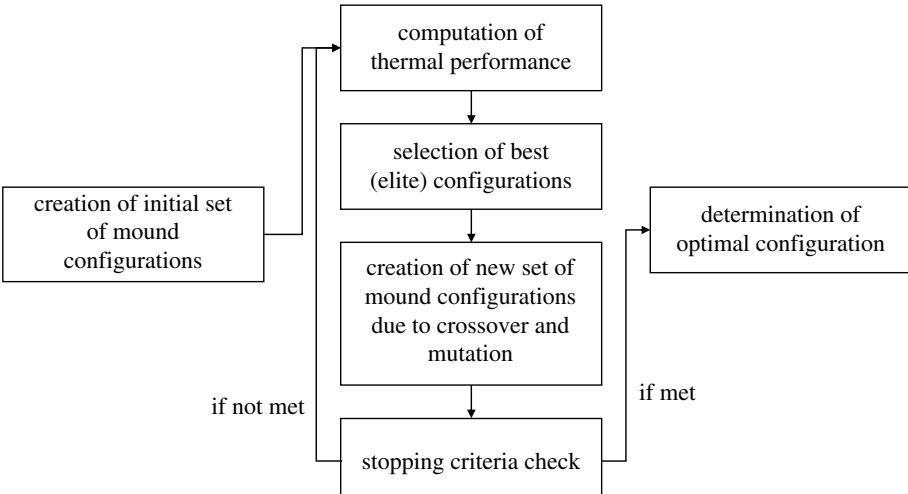

**Figure 12.** Flow chart of the genetic algorithm used in the optimization process.

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
