## [Reviewer comments · Royal Society Open Science]

Review History

RSOS-191332.R0 (Original submission)

Review form: Reviewer 1 (Stuart P. Wilson)

Is the manuscript scientifically sound in its present form?

Yes

Are the interpretations and conclusions justified by the results?

Yes

Is the language acceptable?

Yes

Do you have any ethical concerns with this paper?

No

Have you any concerns about statistical analyses in this paper?

No

Recommendation?

Accept as is

Comments to the Author(s)

I reviewed the original submission (as reviewer 2). The clarity of the manuscript and the scope of the results are much improved through the revisions. The rebuttal letter does a good job of clarifying several issues that I raised. Inclusion of additional simulations without the lateral channels provides some interesting new insights, and addresses many of my original concerns. My only suggestions are i) that the authors consider including the justification for approximating the 3D structure with a 2D model, as represented by figure R1 in the rebuttal, in the paper (perhaps as an appendix), as this is a neat additional result, and ii) *specifically* how the error bars are computed in figures 5-9 should be clarified.

Review form: Reviewer 2 (Jeffrey Scott Turner)**Is the manuscript scientifically sound in its present form?**

Yes

Are the interpretations and conclusions justified by the results?

Yes

Is the language acceptable?

Yes

Do you have any ethical concerns with this paper?

No

Have you any concerns about statistical analyses in this paper?

No

Recommendation?

Accept as is

Comments to the Author(s)

This paper represents an intriguing analysis of Bejan's constructal law as it applies to a major ecosystem engineer – the construction of Macrotermes mounds.

Although I'm not really qualified to judge the models' mathematical details, I see the author has already faced a pretty detailed critique from reviewers who are qualified. As far as I can judge, the author has risen conscientiously and constructively to those criticisms.

I thought the paper was very clearly written, if a bit dense for readers who are not steeped in the mathematics and practice of numerical modeling. My own preference is to relegate the dense material to an appendix and to favor a clear narrative that eases the reader through the paper. That said, I think the author did a very good job of laying out the results, even if the reader had to work hard to see them. I don't think a major revision like this would entail is justifiable at this point. I would recommend just a final editorial scan, but with my comments offered as food for thought in future publications.

I do have a couple of small quibbles about the natural history. There was little attention paid to the behavior of the termites. We've been finding some important differences in behavior that would explain variation of mound architecture. I'm not sure that's important for this paper, which I see first and foremost as an exploration of the constructal law, and so stands on its own without having every detail of the natural history taken into account. I also note that water was not analyzed, even though water dynamics are a very significant factor in shaping the mounds. Mound remodeling, for example, is a process limited to the rainy season, which happens to occur during the period between the summer solstice and autumnal equinox. I don't think that would change the conclusions here, since that is the time of year when the sun is at its average zenith angle. Perhaps just an acknowledgment of the importance of water would be useful, if only as a prompt to consider it in future analyses.

Decision letter (RSOS-191332.R0)

28-Sep-2019

Dear Professor Yaghoobian

On behalf of the Editors, I am pleased to inform you that your Manuscript RSOS-191332 entitled "How the Thermal Environment Shapes the Structure of Termite Mounds" has been accepted for publication in Royal Society Open Science subject to minor revision in accordance with the referee suggestions. Please find the referees' comments at the end of this email.

The reviewers and handling editors have recommended publication, but also suggest some minor revisions to your manuscript. Therefore, I invite you to respond to the comments and revise your manuscript.

- Ethics statement

- Data accessibility

If you wish to submit your supporting data or code to Dryad (<http://datadryad.org/>), or modify your current submission to dryad, please use the following link:
<http://datadryad.org/submit?journalID=RSOS&manu=RSOS-191332>

- Competing interests

- Authors' contributions

- Acknowledgements

- Funding statement

Because the schedule for publication is very tight, it is a condition of publication that you submit the revised version of your manuscript before 07-Oct-2019. Please note that the revision deadline will expire at 00.00am on this date. If you do not think you will be able to meet this date please let me know immediately.

on behalf of Prof Kevin Padian (Subject Editor)
openscience@royalsociety.org

Associate Editor Comments to Author:

The reviewers have provided a few minor suggestions for clarity and to support the direction of future work, and we'd recommend you incorporate these before acceptance. Thanks for the submission.

Reviewer comments to Author:

Reviewer: 1

Comments to the Author(s)

I reviewed the original submission (as reviewer 2). The clarity of the manuscript and the scope of the results are much improved through the revisions. The rebuttal letter does a good job of clarifying several issues that I raised. Inclusion of additional simulations without the lateral channels provides some interesting new insights, and addresses many of my original concerns. My only suggestions are i) that the authors consider including the justification for approximating the 3D structure with a 2D model, as represented by figure R1 in the rebuttal, in the paper (perhaps as an appendix), as this is a neat additional result, and ii) *specifically* how the error bars are computed in figures 5-9 should be clarified.

Reviewer: 2

Comments to the Author(s)

This paper represents an intriguing analysis of Bejan's constructal law as it applies to a major ecosystem engineer – the construction of Macrotermes mounds.

Although I'm not really qualified to judge the models' mathematical details, I see the author has already faced a pretty detailed critique from reviewers who are qualified. As far as I can judge, the author has risen conscientiously and constructively to those criticisms.

I thought the paper was very clearly written, if a bit dense for readers who are not steeped in the mathematics and practice of numerical modeling. My own preference is to relegate the dense material to an appendix and to favor a clear narrative that eases the reader through the paper. That said, I think the author did a very good job of laying out the results, even if the reader had to work hard to see them. I don't think a major revision like this would entail is justifiable at this point. I would recommend just a final editorial scan, but with my comments offered as food for thought in future publications.

I do have a couple of small quibbles about the natural history. There was little attention paid to the behavior of the termites. We've been finding some important differences in behavior that would explain variation of mound architecture. I'm not sure that's important for this paper, which I see first and foremost as an exploration of the constructal law, and so stands on its own without having every detail of the natural history taken into account. I also note that water was not analyzed, even though water dynamics are a very significant factor in shaping the mounds. Mound remodeling, for example, is a process limited to the rainy season, which happens to occur during the period between the summer solstice and autumnal equinox. I don't think that would change the conclusions here, since that is the time of year when the sun is at its average zenith angle. Perhaps just an acknowledgment of the importance of water would be useful, if only as a prompt to consider it in future analyses.

Author's Response to Decision Letter for (RSOS-191332.R0)

See Appendix A.

RSOS-191332.R1 (Revision)

Review form: Reviewer 1 (Stuart P. Wilson)

Is the manuscript scientifically sound in its present form?

Yes

Are the interpretations and conclusions justified by the results?

Yes

Is the language acceptable?

Yes

Do you have any ethical concerns with this paper?

No

Have you any concerns about statistical analyses in this paper?

No

Recommendation?

Accept as is

Comments to the Author(s)

The authors have addressed my concerns to a high standard.

Review form: Reviewer 2 (Jeffrey Scott Turner)

Is the manuscript scientifically sound in its present form?

Yes

Are the interpretations and conclusions justified by the results?

Yes

Is the language acceptable?

Yes

Do you have any ethical concerns with this paper?

No

Have you any concerns about statistical analyses in this paper?

No

Recommendation?

Accept as is

Comments to the Author(s)

Thank you for your response to my comments.

Decision letter (RSOS-191332.R1)

12-Nov-2019

Dear Professor Yaghoobian,

It is a pleasure to accept your manuscript entitled "How the Thermal Environment Shapes the Structure of Termite Mounds" in its current form for publication in Royal Society Open Science. The comments of the reviewer(s) who reviewed your manuscript are included at the foot of this letter.

Kind regards,

on behalf of the Associate Editor, and Professor Kevin Padian (Subject Editor)
openscience@royalsociety.org

Associate Editor Comments to Author:

Thank you kindly for submitting your revision and response to referees' comments. Both Referee #1 and Referee #2 agree that their concerns have been addressed by the authors.

Reviewer comments to Author:

Reviewer: 1

Comments to the Author(s)

The authors have addressed my concerns to a high standard.

Reviewer: 2

Comments to the Author(s)

Thank you for your response to my comments.

Appendix A

We would like to thank the reviewers for taking careful read of the manuscript, which resulted in a much improved paper. Below, we will respond to the specific issues raised by the reviewers.

Referee #1

Comments to the Author

The authors present a solution to a PDE, considering some initial conditions and boundary conditions. They use a straight forward FE approach in arriving at their numerical solution. For extracting any parameters to be used in their study they rely on the book by Turner.

While the authors have clearly mentioned that their numerical exercise is restricted to only the environmental role in assessment of the structure of the termite mound, an important consideration is the structural stability of these soil structures. While it is clear that the environmental considerations are important, as has well been shown by field observations by many researchers and the numerical exercise presented by the authors. It is crucial that the mound structure also emerges due to simple stability considerations.

Response: We agree with the reviewer on the importance of the structural stability of these soil-based structures in their architecture. However, as mentioned by the reviewer, this study focuses on the role of the *thermal* environment on the shape of these structures. Our intent was to give the mound body complete freedom to morph in any direction in response to the thermal environment, assuming that the mounds' construction is strong enough to keep the whole body together under any circumstances. For clarification, this point has been noted in Section 3 (Model description and simulation setup) of the manuscript:

“To achieve this goal, all features of the mound geometry are allowed to freely morph, and it was assumed that the cement (i.e. termite saliva) used in the mounds' construction is strong enough to keep the body together in any circumstances.”

In addition, we have noticed that the title of the paper might be misleading in providing a correct initial perception about the paper content. Therefore, the title of the paper is changed to better reflect the focus of the manuscript. The new title is: “*How the Thermal Environment Shapes the Structure of Termite Mounds*”.

In a recent paper by Singh et al (2019), Science Advances, 5:eaat8520 - a clear conclusions on the porous structure of the wall and its contribution toward thermoregulation is enumerated. While it is instructive to think of the combination of the porosity and thereby the thermoregulation of the mounds, no such considerations are made by the authors in their exercise.

Response: Indeed, porosity is an important factor in the mound function. Our model considers the porosity effect in different ways. The mound superstructure with its internal conduits is a porous structure that enhances permeability, CO₂ diffusivity, and thermal insulation (as also mentioned in Singh et al., 2019). The enhanced permeability and CO₂ diffusivity function is considered in

our model through inclusion of the high conductivity pathways (HCP) in the mound body, representing the higher capacity of the conduits to transfer heat and mass. These details are pointed out in Sect. 3, where we explain the model description. According to the literature, 18% of the mound body, in the mounds built by *Macrotermes michaelseni* species, is composed of a network of tunnels. Therefore, in this work, 18% of the model mound is composed of high conductivity pathways that represent the conduits.

In order to address the thermal insulation function of the porosity, following the reviewer's comment, additional analyses were performed and a new section (Section 5.2) added to the paper. The new analyses address the higher thermal insulation of the body through considering a porosity-based effective thermal conductivity in the model. In general, the mound body (excluding the conduits) is impervious to bulk flow and porous on a microscopic scale that only allows for diffusive exchanges (based on Ocko et al., 2017, and our personal communication with one of the authors, Dr. Hunter King). In Section 5.2, we included new results that reflect the effect of a higher thermal insulation due to the inclusion of porosity in the mound body, which we believe improved the discussion in the paper.

It should be noted that the termite mounds studied in Singh et al., 2019 have very different structures than those investigated in our work. Singh et al., 2019 investigated non-fungus-growing *Trinervitermes geminatus* termite nests, which, unlike mounds of the fungus-cultivating species of *Macrotermes michaelseni*, lack internal tunnels. Due to the difference in the architecture, the function of these two mound types might not be the same.

Another crucial drawback in their study is the 2-D approximation of a 3D structure.

Response: To select a 2D versus a 3D model, we initially speculated that the effect of the external (solar irradiance and wind) and internal (metabolic heat) forces in the third dimension are not vital in capturing the function of the mound. This assumption was mainly due to the observed northward tilt of the mound and its conical shape in nature. In addition, observational studies report that the prevailing wind in the natural location of these mounds is from the south and the dominant solar irradiance is from the north. With this assumption, we tested the results from a 2D model against a 3D model, which we share here.

The two simulations were performed using the same geometrical features and boundary conditions. A comparison between the temperature distribution of the 2D model and a cross-section from the center of the 3D model is shown in Figure R1. The results indicate a strong similarity in the temperature distribution and architecture of the mound between the two models. This similarity indicates that a 2D model can replicate what a 3D model captures, while significantly saves the simulation time.

Figure R1. A comparison between the temperature distribution of the 2D model (right) and a cross-section from the center of the 3D model (left)

The authors also compare the spire-base-ratio, and height-to-base ratio as signatures for comparison. It is not entirely clear why these are specific signatures that can be used as a basis for comparison.

Response: We agree that these measures seem limited. However, the study used any possible geometrical metrics available in the literature (e.g. the mounds' average height, base length, mid-height width, and spire tilt angle) to compare the computational results against the natural mounds' attributes. One reason that we used the geometrical characteristics of the *M. michaelseni* mounds as the base for the simulations was that, unlike for other species, available information on the physical features of these mounds are more detailed in the literature. For a better clarification, the text is improved to reflect the point raised by the reviewer:

“Six degrees of freedom (H/L , L'/L , D_0/L , H_1/H , α , and γ), representing the structural characteristics of the mound, are chosen.”

*“While quantitative data of geometrical characteristics of termite mounds are scarce, in general, Turner (2000) provided measured data on the average height, base width, mid-height width, and spire tilt angle of *M. michaelseni* mounds. These measured data in the dimensionless form defined in this work yield $\tilde{L} = 1.5$, $H/L = 0.9$, $L'/L = 0.423$, and $\gamma = 19^\circ$. These values were used as reference data”*

It is to be noted that the mound structure is dynamics, with constantly evolving structure that the termites are rebuilding. It would be wise to see the limiting cases of such analysis. In that what are the mound shapes reached at extremes of temperatures, and this will perhaps provide an inherent basis for which a comparative structural as well as thermal stability analysis need be carried out.

Response: We agree with the reviewer on this note as extreme conditions may reveal new insights and answers to unknowns. In accord to this, we have increased the maximum range of irradiance ratio to a large value of 150, which represents an irradiance of 1000 W/m² over a mound with 7 W of metabolic heat output. This represents an extreme situation, since 1000 W/m² is the maximum amount of irradiance seen on Earth at the Equator level and 7 W of metabolic heat corresponds to an abnormally small mound (minimum value reported in the literature). All results now reflect this new range. Likewise, the location's latitude, represented by the zenith angle, ranged between 0° and 25°, which encompasses all the regions where termites found on earth. The wind speed also reflects a range of extreme values, as the typical wind speed in *M. michaelseni* termites habitat is of 1.38 m/s and no consistent wind higher than 5 m/s was found in literature.

While the authors also ignore by and large the actual porous structure of the mound as alluded to by several authors including Singh et al, and Ocko et al etc 2017,

Response: This point is addressed in detail in an earlier response above.

Referee #2

This paper presents an interesting computational approach to establishing the particular geometries of termite mounds that optimize nest temperatures under different environmental conditions. Under the assumptions that are made, the model seems to have been formulated according to mostly sensible principles. The paper is generally well written. However, while the challenge of engineering the thermal properties of mound structures is well motivated, the underpinning scientific aims of the study need to be stated more concretely (the question ‘how a mound’s structure emerges to provide a suitable living condition...’, p4, is not at all the question that is addressed, at least as far as the optimality criteria are defined in the simulations). And many details of the model are missing or opaque in the text, making the validity of the approach difficult to evaluate. That said, the central result, that variations in the tilt and width of natural mounds may be thermally optimal given the conditions of the environments in which they are constructed, is interesting.

To convincingly support such a conclusion would require addressing the major issues listed below. These relate firstly to how including lateral conduits in the simulated geometry may mask some more interesting thermal properties of the funelling spire, that would be worth investigating with additional simulations without the lateral conduits. And they relate secondly to a lack of detail in the description of the numerical methods and optimisation process, which presumably can be addressed by modifications to the manuscript text only, albeit substantial edits. Overall, the paper describes a very promising approach, but needs considerable work.

We would like to thank the reviewer for a positive reception of the work. We agree with the reviewer that the main scientific question was not articulated properly. Careful efforts were put into improving the text to better reflect the scientific goal of the study and to better represent the modeling section. In addition, as suggested by the reviewer, additional simulations were performed to investigate the effect of the lateral channels on the mound thermal performance and its architecture. We will respond to the specific issues in detail, below.

Major issues:

1. The authors have optimised the geometric parameters of the simulated mound geometry with respect to one criterion - minimizing the maximum temperature in the nest. This represents an assumption that no specific location in the nest should be too hot, i.e., no specific animals/larvae are at risk of over-heating. But a sensible alternative would be to minimize the average temperature across the nest, i.e., all animals/larvae should not be too hot. Would optimising with respect to the latter criterion lead to different optimal mound geometries, and if so, might the extent to which those differences conform to natural examples support a conclusion about the extent to which these different criteria may have been differentially optimised in nature? It is possible that this issue stems from my misunderstanding about how the key value that is being optimised, θ_{nest} , is arrived upon. It is based on T_{nest} (p11), but how is T_{nest} defined? The average of all mesh elements along a line across the the bottom edge? All elements below the

junction of the chimney and conical base? (see final comments relating to a lack of detail about the numerical integration).

Response: Termite nest within the mound structure is a spherical space located underground, beneath the mound over-ground body. Bottom of the central chimney opens into the nest. In our model, nest locates at the bottom line of the chimney with an opening of D_0 . This is where the metabolic heat, generated by the termites and their symbiotic fungus, inputs into the chimney. The objective function was chosen to be the nest temperature because ‘nest’ is where termite colony resides. In general, the mound over-ground structure is uninhabited and termites found in the over-ground part of the mound are there either for repairing the structure or for defending the colony against intruding predators. Temperature of the nest was obtained by averaging the temperature of the finite element patches at the location of the nest: one third of the base length, for the mound without internal channels, and D_0 (the chimney width), for mounds with internal channels.

For clarity, Fig. 2 of the paper was modified to indicate the location of the nest within the mound. In addition, the text, in several locations, was improved for clarifying the point raised by the reviewer:

*“One important characteristic of *M. michaelseni* mounds is that their over-ground structure is empty and uninhabited, and termites live in an underground spherical-shape nest with an average diameter of 1.5 – 2 (m) [e.g. Darlington, 1987]. Termites found in the over-ground part of the mound are there either for repairing the structure or for defending the colony against intruding predators [Darlington, 1987]”.*

“The mound model is subjected to 1) solar irradiance I_c ($W m^{-2}$) at a zenith angle of χ , 2) a heat rate of q_1 (W) imposed at the nest (located at the center of the mound’s base), representing the colony’s metabolic heat, and 3) the ambient wind.”

“Figure 2. Simplified configuration of typical *M. michaelseni* termite mound with internal channels used in the model. q_1 and I_c , respectively, represent the metabolic heat rate from the nest (illustrated with a hatched patch) and solar irradiance.”

Furthermore, the procedure for calculating the mound temperature is explained in Section 3 as:

“Since in *M. michaelseni* mounds the over-ground structure is uninhabited and termite colony resides in the subterranean nest, the minimization of the nest (rather than the whole body) temperature was selected as the objective function for the optimization process. Numerically, these temperature quantities were obtained by averaging temperature values in the mesh elements located in the heat input zone at the base of the structure”

2. I could find in the text no account for why the ‘funelling angle’ parameter, α , which is free to be as great as 30 degrees (table 1), persists to be zero in all optimal configurations discovered, given that figure 1 clearly shows that natural mounds have a conical spire ($\alpha > 0$). If the conclusion is that the width at the top should be as wide as possible ($\alpha = 0$, given the constraints imposed, table 1), this would suggest that a major property of mound morphology cannot be accounted for by this modelling approach. On page 17 (lines 12-14), the authors suggest that the height of the lateral channels is predicted to be greater than in natural mounds due to this discrepancy. Might removing the lateral channels enable the optimization procedure to instead generate predictions about the thermal implications of a funelling spire? Alternatively, α could be fixed and positive (to represent an assumption that funelling is an inevitable consequence of the nest building process / structural integrity) and then the optimal placement of HI for

different alpha could be determined under these constraints. Either way, this issue needs to be addressed, or at least discussed in some detail (currently not discussed).

Response: We thank the reviewer for pointing out this issue. We believe addressing this point has significantly contributed to the improvement of the paper and revealed new aspects of the mound function.

To address this point we performed additional simulations with the lateral channels removed. The results indicate that, without lateral channels, the optimal mound configuration is strongly influenced by the irradiance ratio, while remains unchanged to the wind and zenith angle parameters. This happens because the central channel, in this case, becomes the main passage to balance the external and internal heat sources. Therefore, the mound configuration reveals a higher sensitivity to the solar irradiance, which was reflected in the mound's architectural attributes, including the funneling angle. The spire funneling occurs to distance the external heat (mainly from the top surface) from the nest, as increasing the funneling angle, as the last resort of the constraint body, yields taller mounds.

In addition, we studied the effects of multiple sets of lateral channels. However, the inclusion of this feature did not affect the mound architecture and did not alter the nest temperature significantly. As such, this feature was not analyzed further. The text were modified in several places to reflect the new analyses:

“A new set of simulations were performed for a mound with only a central chimney to investigate the effect of the removal of the lateral channels on the mound geometry. It is found that the mound optimal configuration remained unaffected to the wind and zenith angle parameters, while significantly influenced by the amount of the irradiance ratio. In contrast to the mound with chimney and lateral channels, the mound without lateral channels is much more sensitive to the intensity of the external heat. That is because, in this case, the chimney is the main passageway to balance the effect of the two heat sources (solar and metabolic), and removal of the lateral channels removes additional effective paths for these heats. As a result, unlike the case with lateral channels, the mound geometry revealed a significant sensitivity to the amount of the irradiance, reflected in the quantities of mound aspect ratio, spire tilt, and funneling angles. The mound aspect ratio varied between 0.6 (for $\tilde{I} = 1$) to 1.1 (for $\tilde{I} > 50$), and the spire tilt angle changed from $\gamma = 1^\circ$ ($\tilde{I} = 1$) to $\gamma = 31^\circ$ ($\tilde{I} > 75$). These large variations appear because, under stronger solar irradiances, taller mounds create greater distances, thus larger resistances, between the nest and the top surface. For higher irradiance ratios, the external heat overwhelms the metabolic heat and becomes more important in increasing the mound temperature. Thus, to lessen the effects of this heat, the mound distances the top surface by becoming taller and more tilted, as there are no lateral channels to effectively diffuse the external heat.

In contrast to the previous configurations, the mound without lateral channels exhibited funneling in the spire for $\tilde{I} > 50$, increasing until it stabilized at $\alpha = 12^\circ$ for $\tilde{I} > 100$. The reason behind the appearance of funneling angle, in this case, is the mound's need for becoming taller under strong solar heats. By increasing the funneling angle, the mound uses the last resource of its constraint body to become taller and separate the top surface further away from the nest. In contrast to the tilting angle, this effect manifests only at high irradiance ratios, indicating that it is more thermally efficient for the mound to tilting its spire than funneling it.

In addition to studying the effects of the removal of the lateral channels, the influence of having several lateral channels with independent heights was also investigated. Inclusion of several lateral channels did not affect the mound architecture and altered the nest temperature compared to those with one set of channels by only $\Delta\theta = 0.01$. Considering the small sensitivity of the model to this additional complexity and the amount of additional computational efforts, mounds with more than one set of lateral channels were not analyzed further. ”

Furthermore, we analyzed the implications of forcing a funneled spire on the mound thermal function. For this, we relaxed the area constraint of the optimal configuration of the mound with lateral channels (seen in Section 4.2). All geometric features were kept the same, other than the funneling angle. The results did not show an effect on the nest temperature ($\Delta\theta = 10^{-4}$) in response to the forced funneling. It can be inferred that the spire may not arise due to thermal processes. The manuscript is modified in Section 5.1.5 to reflect these new analyses:

“To further investigate the absence of spire funneling in the mound optimum configurations, the area constraint was relaxed for the mound with lateral channels and all the aspects of the optimal configuration (Section 4.2), but the funneling angle, were kept unchanged. The results of these investigations indicated that the nest temperature did not alter ($\Delta\theta = 10^{-4}$) in response to the forced funneling. This shows that the configurations previously obtained are not just thermally optimized, but they also have a certain resilience in maintaining their thermal performance. It could be concluded that the spire funneling may not arise due to thermal processes, but rather due to different mechanisms, such as mechanical stability and erosion (not considered in this work).”

3. The parameter H_1 specifies the height of a single chamber assumed to run perpendicular to the axis of the central chimney, representing tunnels that lead to two surface conduits (one on either side of the chimney). I could find no justification for why the height of chambers either side of the chimney should be related by a single parameter H_1 , i.e., assuming that two tunnels connect and are parallel, and no basis for this assumption based on a scan of Turner (2000), to which the authors refer in this regard. Images reported in that paper seem to show multiple conduits (not necessarily two), that are not necessarily connected across the chimney and do not necessarily run perpendicular to it. A more appropriate choice might be to define one tunnel for each conduit, each with an independent height parameter (H_1, H_2, H_3 etc.) and angle. However, I appreciate the additional complexity this might entail, and suggest that adding a representation of the surface conduits is a non-essential complexity to the model, given the potential, explained above, to mask a potential contribution of the spire funelling ($\alpha > 1$). Again, a set of simulations with a central chimney but no lateral conduit would help clarify this issue.

Response: Assuming the same height for both lateral channels was one step towards modeling simplification as our initial investigation revealed no model sensitivity to lateral channels with independent heights. We performed several additional tests, covering a range of extreme and moderate environmental forces. Since the new results indicated no effect on the mound thermal process nor on its architecture, the model set-up was kept the same as before. For clarification, we added the following text to the paper:

“This HCP has two lateral branches of the width of D_1 (m), representing the mound lateral conduits, which protrude from the central chimney at height H_1 (m). The lateral channels on both sides of the chimney were represented by the same height rather than being introduced independently. This simplification towards improving computational efficiency was imposed as our preliminary results indicated no sensitivity to this additional complexity in the model.”

As an example, the optimal configuration obtained for cases with the same and independent lateral channel heights are compared in Figure R2, for $\tilde{I} = 100$. As it is evident from the figure, both set-ups provide similar results.

Figure R2. Temperature distribution and lateral channel layout of the optimal configuration for the same (left) and independent (right) channel heights for $\tilde{I} = 100$, $z = 19$, and $u_0 = 1.38$ m/s.

Selection of several, rather than one, lateral channels on each side did not affect the results, either (see Figure R3). In addition, the width of the lateral channels (D_1) is one of the free parameters in the model that represents the combined effect of multiple channels in a simple way.

Figure R3. Temperature distribution and lateral channel layout over the optimal configurations for singular (left) and double (right) sets of lateral channels for typical environmental conditions of Outjo, Namibia ($\tilde{I} = 14$, $z = 19$, and $u_0 = 1.38$ m/s).

As suggested, additional sets of simulations were performed for cases with no lateral channels, and a new section was added to the paper (explained in more detail in response to point number 2).

4. More detail is required to describe the numerical integration of the finite element model. What software was used? Or will any bespoke code be available as a supplement? As it stands, there is insufficient information / resources for another researcher to verify the results of the numerical integration.

Response: All simulations were performed using the ‘pdeTool’ toolbox of MATLAB. The MATLAB codes developed in this research have been uploaded as part of the electronic supplementary material and are available at <https://datadryad.org/review?doi=doi:10.5061/dryad.b75g05b>. Descriptions and references were added to the manuscript:

“Eq. 1 is solved by applying a finite element method with non-uniform, triangular element, implemented in MATLAB environment, precisely the partial-differential equations (PDE) toolbox [Matlab, 2018].”

“Data accessibility. Codes and materials supporting this article have been uploaded as part of the electronic supplementary material and are available at <https://datadryad.org/review?doi=doi:10.5061/dryad.b75g05b>.”

In addition, we added Appendix 2 in the paper that shows the optimization algorithm flowchart in Figure A1.

Figure A1. Genetic Algorithm Flowchart

5. More detail is also required to explain how the optimisation process is done? Scant reference is made to a genetic algorithm (p12), and while citations to Gosselin and others are given, this paper needs to be understandable in isolation from these references. Is this a population based GA? If not, is the GA run many times from different random initial conditions?

Response: The optimization process was performed with a population-based genetic algorithm. The optimal structure was considered the one with best performance over three independent runs, and every analysis of the results are based on these optimal structures.

For clarifying the optimization process, description of the algorithm, parameters used, and definition of the optimal structures were added to the manuscript, in Section 3.2, as follows:

“For the optimization, a binary, single-objective, elitist genetic algorithm (GA) was utilized, similar to previous work in the literature [Gosselin et al., 2009; da SD Estrada et al., 2015]. The algorithm creates a fixed number (population) of geometries, which are generated randomly, called the first generation (set of individuals). Then, the algorithm calculates the performance of each geometry in respect to the minimization of nest temperature. After that, a portion of the geometries with lowest net temperatures are selected (selection) to generate the population of the next generation (crossover). In the next step, the algorithm has a possibility to update values of the free parameters of individuals from this new population, except in the elite individuals (mutation). Finally, the algorithm checks the stopping criterion, and, if they are not met, creates a new generation, repeating the process. In this work, the algorithm utilized has a population type consisting of a bitstring (binary GA), with populations composed of 60 individuals. The selection was uniform, with 2 elite individuals, 80% of the new population created by crossover (the other 20% are created via mutations). There is a 10% chance for mutation in each degree of freedom of the mutated individuals. The stopping criterion was based on the number of generations without improvement in the performance of the best individual. When the algorithm goes over 20 subsequent generations without improvement, it stops. The algorithm chart is included in Appendix 2.

Results discussed in the following sections are based on the best individual, i.e., the best geometry that arises from the GA optimization.”

In addition, the definition of the optimal configuration was added to the manuscript:

“Additionally, for all numerical analyses performed, the optimum configuration was taken as the one with the best performance (i.e. yielding minimum nest temperature), after the GA was run three times, independently.”

There are error bars in figures 5-9 - are these standard deviations over single sets of optimal parameters from multiple runs, or are they over the final population of sets of parameters generated by a single run?

Response: The error bars show the numerical uncertainty of the results within the resolution of the optimization method.

For clarification, description of the error bars is added to the captions of Figs. 5-9:

“The red line and the error bars respectively represent the overall trend and the numerical uncertainty of the results within the resolution of the optimization method.”

6. Reference is made to results from a ‘gas exchange analysis’ (e.g., p15), but I cannot find a clear description of what this entailed? Was the entire optimisation process repeated against different criteria? All I could find mentioned is that the same dynamics as for heat diffusion were run but for ‘zero scalar concentration’ conditions at the boundary (p11). How the relationship between heat and gas diffusion factored into the optimization criteria is not at all clear.

Response: We agree that this information could lead to a confusion. To account for the gas-exchange function of the mound, the same optimization process was performed considering only the scalar concentration. Since both heat and scalar diffusion processes are governed by the same equation, the dimensionless equation (and the objective function) used in the thermal analyses is valid for gas exchange. The only difference between both analyses was in the boundary condition of the surfaces exposed to the environment, which were set as zero gas concentration in the gas exchange analysis. Results from the gas exchange analysis showed a configuration similar to those obtained in the thermal analysis, with the main difference being in the mound inclination, which was found to be vertically-oriented. For a better clarification, the paper is improved in several places with following texts:

“The diffusion mechanism is the same for heat and scalar and described by the same governing equation. For this reason, the gas exchange process is modeled in a similar fashion to the heat transfer process, but with different criteria. In this case, the boundary condition of the surfaces exposed to the environment are of zero scalar concentration with no external sources. The two heat and scalar concentration optimization processes were performed independently. This additional step was made to investigate whether different criteria for the diffusion process in the dense body of termite mound lead to a different conclusion.”

“The configuration that arose from the gas exchange analysis was alike those obtained from the thermal study, except in the spire alignment, which was found to be oriented vertically. The similarity between configurations shows that the scalar diffusion did not affect the mound configuration, while the contrast in spire behavior provided a substantial evidence of the mound inclination being a consequence of thermal processes within the mound, rather than gas exchange.”

Minor:

7. I notice that on page 11, alpha is omitted from the list in parentheses but included in Table 1. And that D1 is listed in these parentheses but missing from Table 1. As descriptions of ‘the

objective of the analysis' and the free parameters of the optimization, it seems to me that these two lists should be consistent.

Response: α was mistakenly dropped from the list (thanks for pointing this out). It is now added to the list.

Unlike H , L , L' , D_0 , H_1 , α , and γ , H' and D_1 are not defined as the free parameters. Rather, they are determined by the total area and pathway area constraints, respectively, thus are not included in Table 1. A clarification on this point was added to the manuscript:

“Results discussed in the following sections are based on the best individual, i.e., the best geometry that arises from the GA optimization. The range and resolution of each degree of freedom are presented in Table 1 (H' and D_1 are not included, given that they are found by the total area and the pathway area constraints).”

8. I cannot parse the title - it should be revised to be grammatically correct, e.g., The role of the environment in the emergence of termite mound structures. In any case, the emergence/construction of the mounds is not really studied/simulated here - rather, the simulations ask about the criteria against which different structures may be considered optimal, and I think a more appropriate title should reflect this, e.g., 'How the thermal environment shapes the structure of termite mounds'.

Response: We agree with this comment on the inappropriateness of the title. The title has been changed according to the reviewer's suggestion: *“How the Thermal Environment Shapes the Structure of Termite Mounds”*.